# In Vitro Studies of Endophytic Bacteria Isolated from Ginger (*Zingiber officinale*) as Potential Plant-Growth-Promoting and Biocontrol Agents against *Botrytis cinerea* and *Colletotrichum acutatum*

**DOI:** 10.3390/plants12234032

**Published:** 2023-11-30

**Authors:** Alejandro Bódalo, Rogelio Borrego, Carlos Garrido, Hernando José Bolivar-Anillo, Jesús M. Cantoral, María Dolores Vela-Delgado, Victoria E. González-Rodríguez, María Carbú

**Affiliations:** 1Microbiology Laboratory, Department of Biomedicine, Biotechnology and Public Health, Faculty of Marine and Environmental Sciences, University of Cadiz, 11510 Puerto Real, Spain; alejandro.bodalo@uca.es (A.B.); rogelio.borrego@uca.es (R.B.); carlos.garrido@uca.es (C.G.); jesusmanuel.cantoral@uca.es (J.M.C.); 2Programa de Microbiología, Facultad de Ciencias Básicas y Biomédicas, Universidad Simón Bolívar, Barranquilla 080002, Colombia; hernando.bolivar@unisimon.edu.co; 3IFAPA Rancho de la Merced, Sede Chipiona, Camino Esparragosa s/n, 11550 Chipiona, Spain; mdolores.vela@juntadeandalucia.es

**Keywords:** endophytic bacteria, *Zingiber officinale*, *Botrytis cinerea*, *Colletotrichum acutatum*, biocontrol, plant growth promotion

## Abstract

Agriculture currently confronts a multitude of challenges arising from the excessive utilization of chemical pesticides and the proliferation of phytopathogenic fungi strains that exhibit resistance to commonly employed active compounds in the field. *Botrytis cinerea* and *Colletotrichum acutatum* are phytopathogenic fungi that inflict substantial economic losses within agriculture and food due to their high impacts on crops both pre- and post-harvest. Furthermore, the emergence of fungal strains that are resistant to commercial fungicides has exacerbated this problem. To explore more environmentally sustainable alternatives for the control of these pathogens, an investigation into the endophytic bacteria associated with ginger (*Zingiber officinale* Rosc.) was conducted. The primary focus of this study involved evaluating their inhibitory efficacy against the fungi and assessing their potential for promoting plant growth. The endophytic bacteria genera *Lelliottia*, *Lysinibacillus*, *Kocuria*, *Agrococcus*, *Acinetobacter*, *Agrobacterium*, *Zymobacter*, and *Mycolicibacterium* were identified. All isolates showed remarkable in vitro antagonistic ability against *B. cinerea* (>94%) and *C. acutatum* (>74%). Notably, the *Lelliottia amnigena* J29 strain exhibited a notable proficiency in producing extracellular enzymes and indole compounds (IAA), solubilizing phosphate and potassium, and forming biofilm. Furthermore, the *Lysinibacillus capsici* J26, *Agrococcus citreus* J28, and *Mycolicibacterium* sp. J5 strains displayed the capacity for atmospheric nitrogen fixation and siderophore production. These findings underscore the agricultural and biotechnological potential of endophytic bacteria derived from ginger plants and suggest the feasibility of developing alternative approaches to manage these two phytopathogenic fungi.

## 1. Introduction

Modern agriculture faces a multitude of challenges. These include biotic stresses, such as pathogens and pests, as well as abiotic factors like salinity, drought, and temperature extremes. Compounding these issues is the relentless growth of the global population. The cumulative impact of extreme climatic events, ranging from droughts and floods to soaring temperatures, and the unsustainable utilization of water resources have resulted in significant losses in crop production. In addition, agriculture must contend with the loss of biodiversity and the over-application of plant protection products.

The excessive use of agrochemicals is a multifaceted problem with far-reaching consequences, affecting human health [1,2], the environment [3], and the sustainability of agriculture. Within soils, it leads to a decrease in fertility and reduced water retention capacity, ultimately diminishing agricultural yields [4]. Moreover, pesticides have given rise to pests and weeds that have developed resistance to them, posing a formidable challenge to sustainable agriculture [5]. Consequently, recent research endeavors are dedicated to the development of alternative strategies to reduce the dependency on synthetic pesticides [6].

Among the phytopathogenic fungi affecting agricultural crops, *Botrytis cinerea* and *Colletotrichum acutatum* stand out as notorious fungi that are responsible for grey rot and anthracnose, respectively [7,8,9]. Their broad host range includes fruits, vegetables, flowers, and ornamental plants [10]. The damage they inflict not only compromises the quality of agricultural products but also incurs substantial economic losses to farmers and producers. Traditionally, the management of these diseases necessitates multiple fungicide applications throughout the growing season [11]. However, the overuse of fungicides has led to the emergence of chemical-resistant strains [12]. Consequently, the management of these plant pathogens in agriculture necessitates a combination of preventive practices and the targeted use of fungicides [13]. Nonetheless, research is actively exploring biological methods to control these pathogens more sustainably, including genetic enhancements to increase plant resistance and the use of antagonistic microorganisms [14].

Endophytes are microorganisms that reside within plants for at least part of their life cycles without causing apparent harm to their hosts. The relationship between endophytic microorganisms and plants is considered a symbiosis relationship, with both parties deriving benefits from the association. These microorganisms have the potential to play pivotal roles in promoting plant health, shielding against pathogens, and increasing agricultural productivity, making them an important point of sustainable agriculture and biological research. This group of microorganisms includes bacteria, fungi, and viruses, with thousands of different species of endophytes [15]. 

Endophytic bacteria have demonstrated the capacity to fix atmospheric nitrogen in plant roots, thereby improving plant nutrition and reducing the need for nitrogen-based fertilizers. Furthermore, they play a crucial role in promoting plant growth by enhancing nutrient uptake [16]. These bacteria can also act as antagonists to pathogens, competing with them for space and nutrients, producing antimicrobial compounds, or inducing defense responses in plants to protect against diseases [17]. Some of these bacteria assist plants in coping with abiotic stresses, such as drought, salinity, and heavy metal toxicity, by producing enzymes and compounds that facilitate plant detoxification. Additionally, certain endophytic bacteria can aid in the bioremediation of contaminated soils by degrading chemical pollutants within plants [18].

*Zingiber officinale* (ginger) is a perennial plant native to tropical Asia that is cultivated for its subterranean rhizome, and it is widely used as a spice and in traditional medicine for its aromatic and healing properties. Among the most important medicinal properties of this plant are its anti-inflammatory properties, the relief of nausea, the improvement of digestion, and its antioxidant properties [19]. Previous studies have examined the endophytic bacteria in ginger [20,21,22,23] and have demonstrated their potential as plant growth promoters and antifungals. The changes in the population of endophytic bacteria throughout the different growth stages of ginger were analyzed, and it was concluded that fluctuations in the number and type of endophytic microorganisms appear to be influenced by both the plant and the environment [20].

In the context of sustainable agriculture, the main objective of this work was to study the potential of endophytic bacteria isolated from the ginger plant as biocontrol agents against *B. cinerea* and *C. acutatum* and as plant growth promoters.

## 2. Results

### 2.1. Endophytic Bacteria Isolation and Identification

Nineteen endophytic bacterial strains were isolated from *Zingiber officinale* based on the distinctive morphological characteristics of the colonies. Partial sequencing of both the *16S rDNA* and *rpo*β genes was conducted for all isolates. Nucleotide sequencing enabled the identification of eighteen isolates to the species level, while isolate J5 was only identified to the genus level as *Mycolicibacterium* sp. when compared with the NCBI nucleotide database (Table 1). Among the endophytic bacteria isolated from ginger, we identified eight genera and nine different species. The two most predominant genera were *Lelliottia* and *Lysinibacillus*, with five strains assigned to each. The most frequently encountered species among the isolates was *Lelliottia amnigena*, followed by *Lysinibacillus capsici* and *Kocuria polaris*. Additional species included *Lysinibacillus macroides*, *Agrococcus citreus*, *Agrobacterium tumefaciens*, *Acinetobacter schindleri*, and *Zymobacter palmae*.

A neighbor-joining phylogenetic analysis was conducted using the Kimura two-parameter model and a bootstrap test with 5000 runs (MegAlign, DNASTAR^®^ Lasergene v. 7.1.0. package). The sequences of 81 bacterial isolates were retrieved from the GenBank database, representing nine genera and thirty-three species. The phylogenetic tree depicted in Figure 1 was constructed using the 16S rRNA gene sequences of these 81 isolates, encompassing nine genera and thirty-three species. The constructed phylogenetic tree shows the identifications made for each of the 19 isolated bacterial strains, whose identification and accession numbers are provided in Table 1. 

### 2.2. Detection of Genes Involved in the Synthesis of Lipopeptides

The presence or absence of genes encoding for the synthesis of lipopeptides was evaluated via PCR. All of the endophytic ginger bacteria were found to possess at least one gene encoding for the synthesis of a specific type of lipopeptide, except for the *Lelliottia amnigena* J11, *Kocuria polaris* J15, and *Agrococcus citreus* J16 strains (Table 2). *Lelliottia amnigena* J2 and *Agrobacterium tumefaciens* J14 were the strains with the highest number of genes encoding lipopeptide synthesis, which enabled them to produce up to four different types of lipopeptides (bacillomycin, fengycin, iturin, and surfactin) if the conditions were suitable (Table 2). In contrast, *Lysinibacillus capsici* J24, *Kocuria polaris* J17 and J19, and *Acinetobacter schindleri* J30 only possessed one of the six genes encoding the various lipopeptide families. Seven bacterial strains, specifically *Lelliottia amnigena* J12, J21, and J29, *Lysinibacillus capsici* J23 and J26, *Agrococcus citreus* J28, and *Mycolicibacterium* sp. J5, could only synthesize two lipopeptides, subtilin and surfactin. Additionally, *Zymobacter palmae* J20 could produce bacillomycin. The strains with the ability to synthesize three types of lipopeptides included *Lysinibacillus macroides* J22 (bacylisin, iturin, and surfactin) and *Zymobacter palmae* J20 (bacillomycin, subtilin, and surfactin). *Lysinibacillus macroides* J22 is the sole strain harboring the gene responsible for bacylisin synthesis.

### 2.3. In Vitro Antagonistic Activity Assay against Phytopathogenic Fungi

Based on the antagonistic assay using the co-culture method (Figure 2), all endophytic isolates of ginger exhibited an inhibitory capacity against the phytopathogenic fungi *B. cinerea* B05.10 and *C. acutatum* IMI34849 grown on an LB medium (Figure 3), with a more pronounced effect being observed against *B. cinerea*. The % inhibition of *B. cinerea* growth exceeded 94% regardless of the antagonistic bacterial species. However, the inhibitory capacity of ginger endophytes against *C. acutatum* varied and depended on the strain. Among the isolates belonging to the *Lelliottia amnigena* species, J21 and J29 showed the lowest and highest inhibitory activities (74% and 95%; Appendix A) against the *C. acutatum* fungus, respectively. A similar trend was observed among the isolates identified as *Kocuria polaris* (75% and 90% inhibition of the fungus by isolates J15 and J19; Appendix A). Consequently, all of the ginger endophytic bacteria demonstrated their potentials as in vitro biocontrol agents against the phytopathogenic fungi *B. cinerea* and *C. acutatum*. Furthermore, the *Lelliottia amnigena* J29 strain displayed promising in vitro antagonistic activity against both phytopathogens.

### 2.4. In Vitro Characterization of Bacteria for Plant-Growth-Promoting Potential

#### 2.4.1. Extracellular Enzyme Production Assays

The production capacity of hydrolytic enzymes, including amylase, cellulase, esterase, lipase, and protease, was determined for all of the isolates (Table 3). These enzymes are secreted by endophytic bacteria as a defense against pathogens, providing protection to the plant. None of the ginger endophytic strains exhibited amylolytic or cellulolytic activities. However, all isolates showed the ability to produce one or more of the esterase, lipase, or protease enzymes, except for *Lelliottia amnigena* J11 and *Agrococcus citreus* J16 strains, which did not show any enzymatic activity. In contrast, all three activities were observed in the *Lelliottia amnigena* J21 and *Lysinibacillus macroides* J22 strains. Protease activity was observed in the isolates belonging to *Lelliottia amnigena*, *Lysinibacillus capsici*, *Lysinibacillus macroides*, *Kocuria polaris*, *Zymobacter palmae*, and *Mycolicibacterium* sp., representing 53% of the samples. Regarding lipid-degrading enzymes, it was noted that 42% of the strains were capable of excreting both esterases and lipases. These strains belonged to the *Lelliottia amnigena* (J21 and J29), *Lysinibacillus capsici* (J23 and J26), *Lysinibacillus macroides* (J22), *Kocuria polaris* (J15), *Agrococcus citreus* (J28), and *Agrobacterium tumefaciens* (J14) species. Meanwhile, the *Zymobacter palmae* J20, *Acinetobacter schindleri* J30, and *Mycolicibacterium* sp. J5 strains only excreted lipases.

#### 2.4.2. Atmospheric Nitrogen Fixation, Potassium, and Phosphate Solubilization

Nine of the nineteen strains (47.4%) showed nitrogen fixation abilities, belonging to the *Lelliottia amnigena* (J2), *Lysinibacillus capsici* (J24 and J26) and *Lysinibacillus macrolides* (J25), *Kocuria polaris* (J17 and J19), *Agrococcus citreus* (J16 and J28), and *Mycolicibacterium* sp. (J5) species (Table 4). One strain (5.3%), *Lelliottia amnigena* J29, showed phosphate and potassium solubilization abilities (Table 4). The remaining strains (47.3%) were negative for all of the assayed abilities.

#### 2.4.3. Siderophore and IAA Production

The production of siderophores under the test conditions was only possible for four isolates: *Lelliottia amnigena* J12, *Lysinibacillus capsici* J26, *Agrococcus citreus* J28, and *Mycolicibacterium* sp. J5 (Table 4). 

After two days of growth in the “King-B” liquid culture medium under agitation, most of the isolated bacteria synthesized and excreted IAA irrespective of the presence of the precursor in the medium (Table 4). The only strains that did not produce IAA in the absence of tryptophan were *Lelliottia amnigena* J29, *Kocuria polaris* J15, and *Acinetobacter schindleri* J30. The IAA concentrations ranged from 0 to 3.55 µg/mL in the absence of tryptophan, while in the presence of the amino acid, they ranged from 0.07 to 4.6 µg/mL (Figure 4; Appendix A). In the absence of tryptophan, the *Mycolicibacterium* sp. J5 and *Lysinibacillus macroides* J22 strains produced the highest and lowest amounts of IAA, respectively, while in the presence of the precursor, the *Lelliottia amnigena* J29 strain, which did not produce IAA in the absence of the precursor, yielded the highest amount of the acid, and *Acinetobacter schindleri* J30 excreted the lowest amount. The addition of tryptophan to the culture medium resulted in an increase in the amount of IAA excreted by the different isolates except for the *Mycolicibacterium* sp. J5, *Agrococcus citreus* J28, and *Lysinibacillus capsici* J26 strains (Figure 4). The highest increase in production was observed in the *Lelliottia amnigena* J29 strain, followed by the *Agrococcus citreus* J16, *Lelliottia amnigena* J12, and *Lysinibacillus capsici* J24 strains.

#### 2.4.4. Biofilm Production

In general, most of the isolates were able to produce biofilm, except for *Lysinibacillus capsici* J23, *Lysinibacillus macroides* J22, *Kocuria polaris* J15, *Zymobacter palmae* J20, and *Acinetobacter schindleri* J30. Based on the classification criteria described by Stepanovic S. et al. [24], the analyzed endophytic bacteria were classified into strongly adherent, moderately adherent, or weakly adherent strains (Table 5). Among the strongly adherent strains were *Lysinibacillus capsici* J26, *Kocuria polaris* J19, *Agrococcus citreus* J28, and *Mycolicibacterium* sp. J5, while *Kocuria polaris* J15, *Zymobacter palmae* J20, *Lysinibacillus macroides* J22, *Lysinibacillus capsici* J23, and *Acinetobacter schindleri* J30 were classified as not adherent strains.

## 3. Discussion

The agricultural sector faces pressing challenges such as increasing food demand, climate change, and the urgent need to ensure plant health. Existing agricultural methodologies, particularly in the field of crop protection against pathogen infection, are becoming environmentally obsolete. The scientific community has been searching for more sustainable, efficient, and specific alternatives for combating fungal pathogens, among others. A promising alternative is the use of endophytic microorganisms with a high potential for producing antimicrobial compounds, demonstrating effective antagonism upon direct contact with the pathogen. Endophytic bacteria in general, and the bacteria isolated in this study in particular, were shown to have the abilities to control *B. cinerea* and *C. acutatum*, which, as seen a priori and in the in vitro assays conducted, are very promising and could represent an effective option for future phytopathogen control.

In this study, 19 endophytic bacteria isolated from ginger rhizomes were identified as *Lelliottia*, *Lysinibacillus*, *Kocuria*, *Agrococcus*, *Acinetobacter*, *Agrobacterium*, *Zymobacter*, and *Mycolicibacterium*. In many cases, the presence of one genus or another of bacteria in symbiosis with a plant species depends, to some extent, on the metabolic compatibility between the organisms, as well as on the presence of both in the same microhabitat at a certain point in time. Previous studies isolated bacteria in ginger belonging to some of the genera isolated in our work, specifically *Lelliottia*, *Enterobacter*, *Acinetobacter*, and *Agrobacterium* [20,25]. Factors such as the plant age, the tissue analyzed, the plant genotype, soil fertility and texture, and the timing of the plant sample collection and soil management methods have been described as factors that can influence the composition of endophyte communities [25]. However, it was concluded that the ginger rhizome can provide a stable niche for specific communities to thrive, and therefore, there must be some selectivity among the possibilities of establishing a relationship between them [26].

At the species level, nine bacterial isolates were identified as *Lelliottia amnigena*, which is one of the main bacteria that was characterized as an endophyte in ginger in the literature [27]. It is important to note that *Lelliottia amnigena* has also been isolated from other plants, such as the phanerogam *Euphorbia prostrata* [27] and the roots of *Zea mays* [28]. Other genera and species identified in this work, such as *Agrobacterium tumefaciens*, have been previously published as endophytic bacteria isolated from the root parts and stems of leguminous plants, such as *Onobrychis viciaefolia* [29], and in the herbaceous plant *Oxalis corniculata* [30]. Another important group of isolated bacteria comprises strains from the Actinobacteria phylum, including *Arthrobacter*, *Kocuria*, and *Agrococcus*, which are well-known endophytic bacteria in numerous plants [31,32]. Specifically, *Kocuria polaris* has been isolated from *Scrophularia striata* [33]. *Lysinibacillus macroides*, which was previously described as an endophytic bacterium, has been isolated from wheat roots [34] and *Paspalum vaginatum* [35]. *Zymobacter palmae*, on the other hand, has only been described as an endophyte in palm sap [36]. Several strains of *Acinetobacter schindleri* have been described as endophytic bacteria, including those associated with *Pseudostellaria heterophylla*, a traditional Chinese medicinal plant [37], and in the roots of various plants [38]. Furthermore, species of the genus *Mycolicibacterium* have been documented as endophytic microorganisms in various plants, including potato roots (*Solanum tuberosum*) [39] and coffee roots (*Coffea canephora* and *C. liberica*) [40]. Bacteria belonging to these genera, such as *Agrococcus baldri*, have previously been identified in the roots, stems, and leaves of *Vitis vinifera* [41]. Finally, it is worth noting that the *Agrococcus citreus* and *Lysinibacillus capsici* species are described for the first time in this study as potential endophytic microorganisms in plant root parts, and specifically, they had never been isolated from ginger before.

The characterization study of antimicrobial activity, carried out in direct in vitro confrontational cultures, has yielded very promising results in practically all of the isolated and identified genera. The antifungal control results were greater against the *B. cinerea* pathogen, with inhibition levels exceeding 90%. Meanwhile, for the *C. acutatum* fungus, protection levels ranging from 70% to 90% inhibition were achieved. Despite the scientific advances that were previously achieved in other scientific studies, the resistance exhibited by some strains of *Botrytis* continues to cause damage and economic losses [42]. Consequently, there is an urgent need for the development of innovative strategies to control *B. cinerea* infection in agriculture, including preventive measures [43]. For example, a biofungicide containing the *Bacillus subtilis* bacterium was applied prophylactically in vineyards (*Vitis vinifera* cv. Tempranillo), and its effects were compared to those of a chemical fungicide composed of fenhexamid in terms of oenological parameters [44]. The results showed that the application of *Bacillus subtilis* did not compromise the quality of the grapes or wine, and it provided protective effects like those of the chemical fungicides, positively influencing grape production in the vineyards. These findings underscore the viability and environmental friendliness of biofungicides as a strategy for gray mold control in vineyards, especially where grapes are susceptible to *Botrytis* infections, without interfering with oenological parameters.

On the other hand, the biocontrol capacity of *Colletotrichum* has been investigated in previous studies. It has been demonstrated that endophytic bacteria from the bean plant inhibit the growth of the *Colletotrichum indemuthianum* fungus in vitro. For example, strains of *Bacillus subtilis* inhibited the growth of *C. indemuthianum* by 96.96%, while isolates of *Streptomyces cyaneofuscatus* and *Streptomyces flavofuscusotras* inhibited its growth by 75.55% and 79.99%, respectively [45]. Onion endophyte isolates have also shown an inhibitory capacity against the growth of *Colletotrichum gloeosporioides*, with inhibition percentages ranging from 33.3% to 73.3% in in vitro assays [46]. Given the significant degree of in vitro inhibition demonstrated by all strains against *B. cinerea* B05.10 and *C. acutatum* IMI34849, our results are very promising and are above those shown by other authors in a similar line of work. In view of the results presented in this work, the bacteria isolated from ginger produce some type of compound or mixture of compounds with marked antifungal properties, and in the future, with further attention and characterization of these isolates at the metabolomic level, they could represent a highly effective alternative for the control of these phytopathogens.

In the field of study and in the characterization of endophytic microorganisms, specifically applied to agri-food, there is the possibility that these microorganisms may have a very positive influence on plant growth promotion. Previous studies have shown that endophytic bacteria can have a positive impact on plant growth and development, influencing processes such as nitrogen fixation, phosphate, and potassium solubilization, the production of phytohormones like indoleacetic acid (IAA), and siderophores [47]. In the present study, a phenotypic and biochemical characterization of our isolates was carried out in relation to a series of characteristics that could make them very interesting for use as plant growth promoters in agricultural crops.

Endophytic bacteria have previously been an important source of extracellular enzymes, capable of degrading organic compounds such as cellulose, proteins, carbohydrates, and lipids. These enzymes can play crucial roles in improving nutrient absorption by plants, soil fertility enhancement, and the potential reduction in the dependence on synthetic fertilizers and pesticides [48,49,50]. In this study, a qualitative evaluation of the production capacity of five hydrolytic enzymes by ginger endophytic bacteria was conducted. Among the strains studied, *Lysinibacillus macroides* J22 and *Lelliottia amnigena* J21 stood out for their abilities to produce esterase, lipase, and protease enzymes. Recent research indicates that endophytic bacteria tend to prominently produce lipases and esterases compared to other enzymes, which is attributed to their adaptation to the metabolic environment of the host tissue and environmental conditions [51]. Furthermore, it has been documented that the lipolytic activity of endophytic bacteria, particularly the production of lipase and esterase, is crucial due to their involvement in the hydrolysis of major components of the cell walls of phytopathogenic fungi [17]. This activity has been associated with strains with strong antimicrobial properties [52]. It is important to note that *Lysinibacillus macroides* has previously been identified as a producer of antimicrobial compounds and enzymes such as chitinase, glucanase, and protease, which inhibit fungal hyphae development by degrading cell walls [53,54]. In contrast, recent isolates of *Lelliottia amnigena* did not exhibit enzymatic activities such as pectinase, lipase, cellulase, and amylase [27].

In the context of this study, the in vitro capacity of the 19 ginger endophytic bacteria to fix atmospheric nitrogen was evaluated, which has been described as one of the potential positive effects exhibited by endophytic microorganisms. In our study, more than 50% of the ginger endophytic strains, specifically those belonging to the genera *Lysinibacillus*, *Kocuria*, *Agrococcus*, and *Mycolicibacterium*, had the ability to fix atmospheric nitrogen. Furthermore, it was observed that the *Lelliottia amnigena* J29 strain had the ability to solubilize phosphate and potassium. Nitrogen, phosphate, and potassium are essential for plant growth due to their key roles in physiological and metabolic processes [55,56]. Therefore, maintaining adequate levels of these nutrients in the soil is essential for healthy and optimal plant growth and development [57]. This is particularly important given the adverse environmental implications associated with excessive reliance on synthetic fertilizers. In a previous study, Parashar et al. [27] validated the plant-growth-promoting effects of two *Lelliottia amnigena* isolates. These isolates showed significant improvements in the growth and productivity of wheat and tomato crops under ex vitro conditions. Furthermore, improvements in the key physiological parameters, such as the chlorophyll levels, carotenoids, phenols, and flavonoids, were observed, suggesting comprehensive enhancements in plant stress resistance and physiological vigor.

Another important parameter associated with plant growth promotion in previous studies is the production of indole-3-acetic acid (IAA), a phytohormone that regulates a wide range of plant growth and development processes, including cell elongation, root formation, flowering, fruiting, stomatal regulation, responses to gravity and light, and plant propagation [58]. Its ability to influence these processes is critical to plant success in their environment and their ability to adapt to changing and adverse conditions. Two main pathways for IAA biosynthesis have been proposed, independent of tryptophan and tryptophan-dependent pathways [59]. In this study, it was observed that all ginger endophytic strains produced IAA in the presence of tryptophan and, except for two strains, also in its absence. These results suggest that ginger endophytic bacteria could possess significant potential as biofertilizers [27]. Moreover, the ability to produce IAA in the absence of tryptophan suggests that these bacteria may utilize alternative auxin biosynthesis pathways [60,61], making them more versatile and adaptable to different environmental conditions.

The availability of metallic ions, particularly iron, has also been described as a fundamental parameter in plant development. Under stress conditions, siderophore synthesis is one of the key mechanisms that bacteria use to supply easily available forms of iron to plants [62]. This process not only plays a significant role in the competition with pathogens, but also enhances disease resistance and facilitates beneficial symbiosis with nitrogen-fixing bacteria [63]. In the conditions of the assay conducted in this study, the *Lelliottia amnigena* J12, *Lysinibacillus capsici* J26, *Agrococcus citreus* J28, and *Mycolicibacterium* sp. J5 strains demonstrated the ability to produce siderophores. This could allow them to reduce the availability of iron ions for their competitors, as previously demonstrated [64,65].

Another interesting quality described in endophytic bacteria is the ability to form biofilm. The formation of biofilm by bacteria not only enhances bacterial survival but also contributes to plant growth through various mechanisms [66]. These mechanisms include the biological control of pathogenic organisms, the competitive colonization of plants, and the production of antimicrobial compounds [67,68,69]. When the ability of the isolated bacteria in our study to form biofilms was analyzed, it was observed that 73.7% of the ginger endophytic strains exhibited the ability to produce biofilm, with some of them classified as strong adherents. Furthermore, bacteria that are capable of forming biofilm have shown increased ammonia production, IAA production, phosphate solubilization, siderophore production, and/or nitrogenase activity compared to non-biofilm-forming inoculants [67,70,71,72]. In this study, the isolates that are capable of forming biofilm also performed other crucial functions that support plant development, including atmospheric nitrogen fixation, phosphate and potassium solubilization, siderophore production, and IAA synthesis (Table 5).

In our study, we also analyzed an interesting feature possessed by certain endophytic bacteria, which may play a role in promoting plant growth, and to some extent, act as a mechanism for pathogen control. In this regard, lipopeptides produced by beneficial bacteria play a critical role in promoting plant growth by suppressing pathogens, inducing host defense responses, improving nutrient absorption, stimulating root growth, and contributing to the overall soil health [73,74]. According to their biological activity, lipopeptides are classified as antimicrobial agents, surfactants, and plant growth promoters. The detection of genes involved in lipopeptide biosynthesis in our ginger isolates revealed that most strains had the ability to synthesize the lipopeptide surfactin, with a significant portion also capable of producing subtilin. Only a few strains demonstrated the ability to synthesize other lipopeptides. Our findings are consistent with previous observations, suggesting that these metabolites play a fundamental role in the plant–environment competition [75].

Among the described lipopeptides, surfactins have been noted for their antimicrobial capacity. This antimicrobial activity of surfactins is attributed to their ability to integrate into the lipid bilayers of cell membranes and disrupt their integrity. This is particularly effective against membranes with a low sterol content, which is associated with the fungitoxicity of surfactins [74]. Vine seedlings exposed to surfactins and subtilin exhibited increased resistance to infection caused by *B. cinerea*. These metabolites were found to activate defense genes in the plant, mitigating the damage caused by the fungus [76]. Lipopeptides were isolated from *Bacillus subtilis* strains that demonstrated the ability to produce these compounds, and their presence was directly correlated with each strain’s ability to inhibit the mycelial growth of *B. cinerea* [77]. Additionally, a significant amount of the lipopeptide subtilin produced by *Bacillus subtilis* showed an antagonistic effect on *Candida* spp. [78]. The presence of genes associated with these metabolites in our strains suggests their potential as possible antifungal biocontrol agents.

## 4. Materials and Methods

### 4.1. Isolation of Endophytic Bacteria from Ginger

Ginger (*Zingiber officinale*) plants used in this experiment were collected from an experimental field at IFAPA (Instituto de Investigación y Formación Agraria, Pesquera, Alimentaria y de la Producción Ecológica) located in Chipiona (Cádiz, Spain) in 2019. The plants were transported to the laboratory in sterile cold packaging and processed immediately.

Ginger rhizome samples (250 to 500 g) were washed with sterile distilled water to remove soil, and the surfaces were sterilized with 1% NaClO for 3–5 min. The tissue was then washed several times with sterile distilled water. Surface sterilization was confirmed by the absence of bacterial growth on LB Agar (Miller) medium (Scharlau, Barcelona, Spain) plates inoculated with aliquots of the final wash solution. The superficial bark of the tuber was removed with a sterile scalpel. Then, the samples were crushed with a sterile mortar and pestle and macerated in a 0.9% NaCl solution. A 90 µL suspension was spread on LB medium plates and incubated at 25 °C for 48 h. The bacteria were selected based on the morphology of the colonies and isolated on plates with LB agar medium. 

### 4.2. Molecular Identification of Ginger Isolates

Genomic DNA was extracted following the protocol described by González-Rodríguez et al. [79]. Two pairs of primers were used for partial amplification of *16S* rRNA and *rpo*β genes: 16SF-16SR and Univ_rpoβ_F-R (Table 6) [80,81].

PCR amplifications were performed in a SimpliAmp Thermal Cycler (Applied Biosystems, Foster City, CA, USA) as follows: a total volume of 50 µL containing 0.5 µg template DNA with 1× green Go Taq^®^ Flexi buffer, 2.5 mM MgCl_2_, 0.2 mM of each dNTPs, 0.2 µM of each primer, and 1.25 u GoTaq^®^ G2 Flexi DNA Polymerase (Promega, Madrid, Spain). Cycling conditions were as follows: (a) 16SF-16SR: 95 °C for 2 min, 30 cycles of 95 °C for 0.5 min, 63 °C for 0.5 min, and 72 °C for 1 min, and a final extension step at 72 °C for 10 min; (b) Univ_rpoβ_F-R: 95 °C for 2 min, 30 cycles of 95 °C for 1 min, 60 °C for 1 min, and 72 °C for 1 min, and a final extension step at 72 °C for 10 min. Gel electrophoresis separations were performed using standard procedures [80], and products were purified using the GeneJET PCR Purification Kit (Thermo Scientific, Madrid, Spain).

The products were purified with the GeneJET PCR Purification Kit (Thermo Scientific) and sent to Macrogen for sequencing. Sequences were assembled using the DNASTAR^®^ Lasergene package (DNASTAR, Inc., Madison, WI, USA), and complementary strands were compared using the Basic Local Alignment Search Tool (BLAST) with the nucleotide database from the National Centre for Biotechnology Information (NCBI). Nucleotide sequences were deposited in GenBank (http://www.ncbi.nlm.nih.gov/Genbank/; accessed on 24 November 2023; accession numbers are shown in Table 1). Sequences were aligned, and a neighbor-joining phylogenetic analysis was conducted using MegAlign from the DNASTAR^®^ Lasergene package (DNASTAR, Inc., Madison, MI, USA). To study the phylogenetic relationship of our isolates, eighty-one sequences of related genera and species were downloaded from the GenBank database and included in the phylogenetic tree (Figure 1).

### 4.3. Detection of Genes Involved in the Synthesis of Lipopeptides

A study on genes involved in bacterial lipopeptide pathways was conducted by Mora et al. [75], in which they identified a total of six genes. They also designed specific PCR primers for the partial amplification of each of these genes, namely *itu*C, *fen*D, *bac*A, *sfr*AA, *spa*S, and *bmy*B (Table 7) [75,77]. In our research, we aimed to investigate the presence of these six genes within the genomes of our bacterial isolates. PCR amplifications were run at a total volume of 50 containing 1× buffer, 1.5 mM MgCl_2_, 0.2 mM dNTP, 0.2 µM of each primer, 2.0 U of GoTaq^®^ G2 Flexi DNA Polymerase (Promega), and 0.5 µg of genomic DNA. The cycling conditions were as follows: 95 °C for 5 min, 40 cycles of 94 °C for 1 min, annealing temperature for 1 min, and 72 °C for 1 min. A final extension step at 72 °C for 10 min was followed by a 4 °C soak. The annealing temperature was set, as described by Bolivar-Anillo et al. [77], to 58 °C for *fen*D, *itu*C, *sfr*AA, *bac*A, and *spa*S, and to 55 °C for *bmy*B. PCR products were separated via gel electrophoresis using standard procedures [75,77,80].

### 4.4. In Vitro Antagonistic Activity Assay against Phytopathogenic Fungi

The antifungal potential of the isolated bacterial strains against *B. cinerea* and *C. acutatum* was assessed in vitro using the co-culture method [77,82]. In this study, *B. cinerea* B05.10 and *C. acutatum* IMI34849, which were isolated from grapevines and strawberries, respectively, were selected as the plant pathogenic fungal strains. The phytopathogenic fungi were pre-cultured for seven days at 25 °C in PDA medium, while the endophytic bacteria were cultured for two days at the same temperature in liquid LB medium. For co-culture, bacterial strains were inoculated at a concentration of 1–10^5^ cells/mL in LB medium from liquid medium, placed about 3 cm from a 5 mm mycelial disk of *B. cinerea* or *C. acutatum* taken from solid PDA medium. The antagonistic assays were incubated at 25 °C for a duration of seven days. All bacterial isolates were subjected to evaluation in three independent replicates. As a positive control, Petri dishes were inoculated only with mycelial discs of each fungal strain. The antagonistic effect was calculated using the method described by Tenorio-Salgado et al. [83], where *R_c_* represents the mean radius of fungal growth in the absence of bacteria, and *R* signifies the radius of fungal growth in the presence of antagonistic bacteria.
$$Radial\;inhibition\;\left(\%\right)=\frac{R_{c}-R}{R_{c}}\times 100$$

### 4.5. In Vitro Characterization of Bacteria for Plant Growth Promoting Potential

Each bacterial strain was cultured in liquid LB medium for 24 h at 25 °C. Fresh cultures were then centrifuged at 8963× *g* for 10 min at 4 °C, and the resulting pellet was suspended with 100 µL of LB medium for further assays. All of the following experiments were conducted in triplicate.

#### 4.5.1. Extracellular Enzyme Production Assays

The capacity of endophytic bacterial strains from ginger to secrete enzymes, including amylase, cellulase, esterase, lipase, and protease, was investigated [84]. The cell suspension was then evenly spread on specific agar media using a sterile culture loop, forming a line on the surface to assess the secretion of each enzyme.

To evaluate amylase activity, the bacterial strains were subcultured on Starch agar plates (5% tryptone soy agar (TSA) medium (Oxoid) supplemented with 1% soluble starch). After 72 h of incubation at 25 °C, the plates were stained with Lugol’s solution. The presence of a transparent halo surrounding the colonies indicated a positive result.

For the assessment of cellulase activity, the strains were subcultured in a culture medium composed of 1% NaCl, 1% tryptone, and 0.5% yeast extract supplemented with 0.5% (*w*/*v*) sodium carboxymethylcellulose. After 72 h of incubation at 25 °C, the appearance of a transparent halo around the colonies signified positive cellulolytic activity.

To determine esterase activity, the bacterial strains were subcultured on Tween 80 agar (containing 1% bacteriological peptone, 1% (*v*/*v*) Tween 80, 0.5% NaCl, and 0.01% CaCl_2_·2H_2_O at pH 7.4) and incubated for 120 h at 25 °C. The formation of a white precipitation halo around the colonies was indicative of esterase activity.

Lipolytic activity was detected by the subculture of each strain on a culture medium composed of 1% bacteriological peptone, 1% (*v*/*v*) Tween 20, 0.5% NaCl, and 0.01% CaCl_2_·2H_2_O at pH 7.4. After 120 h of incubation at 25 °C, a precipitate around the colonies was considered positive activity.

To detect protease activity, the bacterial strains were subcultured in skim milk medium (comprising 10% skimmed milk, 0.5% tryptone, 0.25% NaCl, 0.25% yeast extract, and 0.1% glucose at pH 7.0) and incubated at 25 °C for 72 h. The presence of a clear, unformed degradation zone surrounding the colonies indicated proteolytic activity. All assays were conducted in triplicate.

#### 4.5.2. Atmospheric Nitrogen Fixation

To study the strains’ capacity to use atmospheric nitrogen for growth, a semi-solid JMV culture medium (composed of 0.5% mannitol, 0.18% KH_2_PO_4_, 0.06% K_2_HPO_4_, 0.02% MgSO_4_·7H_2_O, 0.01% NaCl, 0.005% yeast extract, and 0.002% CaCl_2_·2H_2_O at pH 5.5–5.7) was inoculated with each endophytic bacterium and then incubated at 25 °C for 7 days [85]. The observation of bacterial growth was considered a positive result.

#### 4.5.3. Potassium Solubilization

The Aleksandrov culture medium (comprising 0.5% dextrose, 0.2% CaPO_4_, 0.2% KAlSi_3_O_8_, 0.05% MgSO_4_, 0.01% CaCO_3_, and 0.0005% FeCl_3_ at pH 7.0–7.5) was employed to assess the capacity to solubilize potassium [26]. The endophytic bacteria were inoculated, and plates were incubated at 25 °C for 3 days. The formation of a clear halo around the colony was considered a positive result.

#### 4.5.4. Phosphate Solubilization

To assess the ability of each strain to solubilize phosphate, we utilized Pikovskaya’s culture medium (composed of 1% dextrose, 0.5% Ca_3_(PO_4_)_2_, 0.05% (NH_4_)_2_SO_4_, 0.05% yeast extract, 0.02% KCl, 0.01% MgSO_4_, 0.00001% FeSO_4_, and 0.00001% MnSO_4_ at pH 7.0–7.4) [86]. The endophytic bacteria were inoculated, and plates were incubated at 25 °C for 10 days. The presence of a clear halo around the colony indicated a positive result.

#### 4.5.5. Siderophore Production

The endophytic bacteria were inoculated in “King-B” culture medium (consisting of 2% peptone, 1.5% (*v*/*v*) glycerol, 0.15% K_2_HPO_4_, and 0.15% MgSO_4_·7 H_2_O at pH 7.0) and incubated for 7 days with shaking at 25 °C. Afterward, the culture media were centrifuged at 14,938× *g* for 10 min at 4 °C, and 100 µL of the supernatant was mixed with an equal volume of 0.1 mM Chrome Azurol S (CAS) solution (30.24 g/L of piperazine-N,N′-bis(2-ethanesulfonic acid) (PIPES), 72.8 mg/L of hexadecyltrimethylammonium bromide (HDTMA), 60.5 mg/L CAS, and 2.7 mg/L FeCl_3_ in 10 mM HCl, pH 5.6) [87]. The mixture was incubated at room temperature for 3 h. A change in the color to yellow indicated the production of siderophores. Commercial siderophore deferoxamine mesylate (DFOM, Sigma-Aldrich, Burlington, MA, USA) was added to “King B” medium as a positive control.

#### 4.5.6. Indole Acetic Acid (IAA) Production

The bacteria were grown on “King-B” medium with and without the supplementation of 500 mg/L L-tryptophan and incubated with shaking for 48 h at 25 °C [77]. The cultures were then centrifuged at 8963× *g* for 5 min at 4 °C. Equal volumes of supernatant and Salkowsky’s reagent were mixed and incubated in the dark at room temperature for 30 min in a multiwell plate. The absorbance was measured at a wavelength of 530 nm.

To determine the concentration of IAA produced by each strain under both conditions, an IAA standard curve (Sigma-Aldrich) was prepared within the range of 0 to 60 µg/mL.

#### 4.5.7. Biofilm Production

Endophytic bacteria were inoculated into LB culture medium and incubated for 24 h at 25 °C. After this time, the optical density of the cultures was measured and then diluted to an OD600 of 0.3. Then, 5 µL of this culture was added to 195 µL of LB medium in a multiwell plate and incubated at 25 °C for 12, 18, and 24 h. The culture medium was carefully removed from the wells, and each well was washed by immersing with sterile distilled water. Then, 150 µL of a 1% crystal violet solution was added and incubated for 30 min at room temperature. The wells were washed twice with sterile distilled water. To quantitate biofilm formation, 150 µL of 33% acetic acid was added to each well, and its absorbance at a wavelength of 570 nm was measured using the MultiSkan FC plate reader (Thermo Scientific) [88]. LB medium without inoculation of any bacteria was used as a negative control.

Strains were classified according to the optical density (OD) values of bacterial biofilms using the classification criteria described by Stepanovic S. et al. [24]. The categories were defined as follows: non-adherent (OD ≤ ODc), weakly adherent (ODc < OD ≤ 2xODc), moderately adherent (2xODc < OD ≤ 4xODc), or strongly adherent (4xODc < OD), where the cutoff OD (ODc) is defined as three standard deviations above the mean OD of the negative control.

### 4.6. Statistical Analysis

The data were statistically analyzed using GraphPad Prism (Version 7.03 for Windows, GraphPad Software, Boston, MA, USA). One-way analysis of variance (ANOVA) was used for multiple sample comparison, followed by Tukey’s HSD test at *p* ≤ 0.05.

## 5. Conclusions

The agricultural sector faces pressing challenges stemming from the overuse of agrochemicals and the emergence of fungal phytopathogens that are resistant to conventional pesticides. This study isolated nineteen endophytic bacterial strains from ginger rhizomes, identifying promising genera including *Lelliottia*, *Lysinibacillus*, and *Kocuria*. All isolates exhibited remarkable in vitro antagonism against *Botrytis cinerea* and *Colletotrichum acutatum*, with inhibitions exceeding 94% and 74%, respectively. These phytopathogenic fungi inflict considerable economic losses, underscoring the need for innovative and sustainable disease management strategies. The ginger endophytes also displayed diverse plant-growth-promoting capabilities, including hydrolytic enzyme production, atmospheric nitrogen fixation, phosphate solubilization, IAA synthesis, and biofilm formation. Notably, *Lelliottia amnigena* J29 showed proficiency in these areas and strong antifungal effects. The presence of genes linked to beneficial lipopeptide biosynthesis further highlights the biotechnological potential of these bacteria.

Ginger endophytes show promise as biocontrol agents and plant growth promoters, with the potential to reduce reliance on chemical inputs and promote sustainable agriculture. Field trials are required to validate their efficacy in establishing effective symbioses with crops and exerting beneficial effects on overall plant health, productivity, and stress resilience. This would pave the way for their commercial development as biopesticides and biofertilizers. This study highlights the potential of ginger endophytes as valuable tools for environmentally friendly agriculture.

## Figures and Tables

**Figure 1 plants-12-04032-f001:**
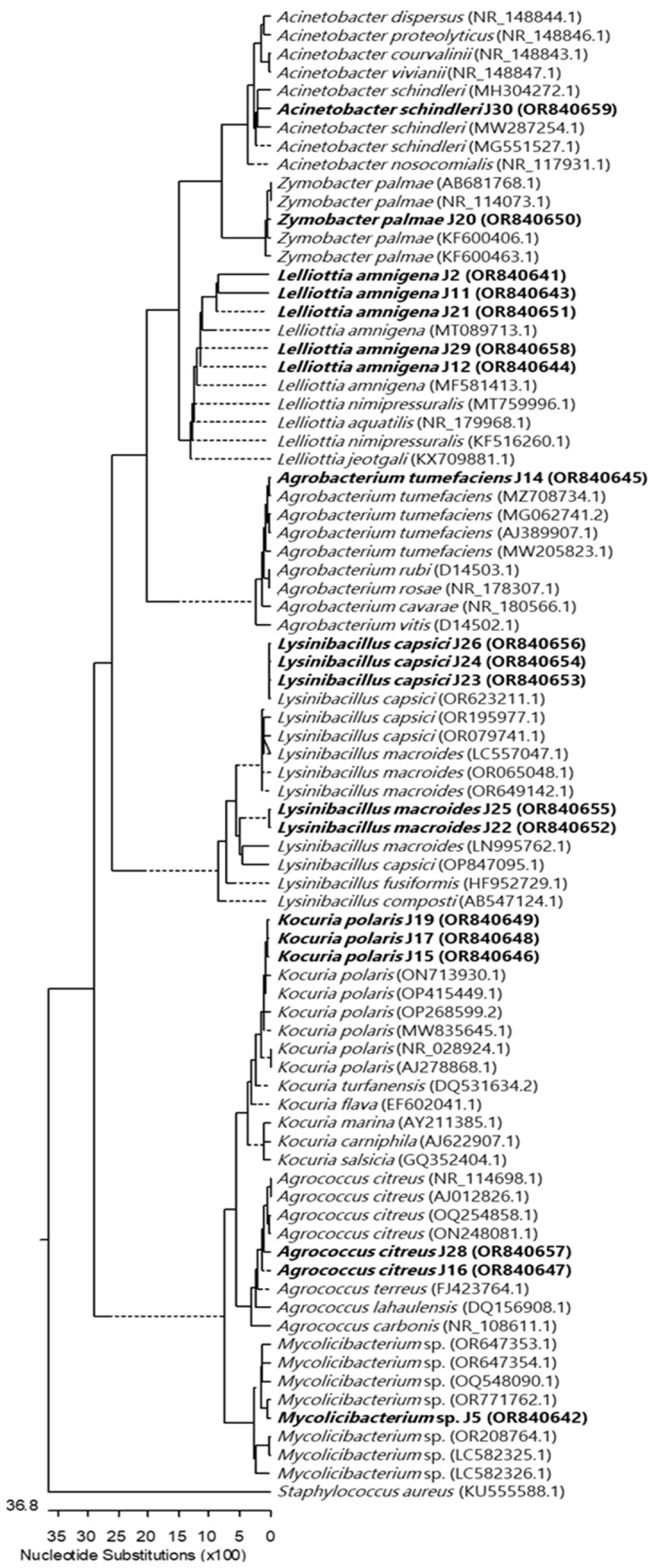
Neighbor-joining tree constructed using 16S rRNA gene sequences, comprising sequences identified in this study (highlighted in bold) and published sequences obtained from the GenBank database. The length of each branch pair reflects the distance between respective sequence pairs. A dotted line on the tree denotes a negative branch length, while the bar indicates the number of nucleotide substitutions.

**Figure 2 plants-12-04032-f002:**
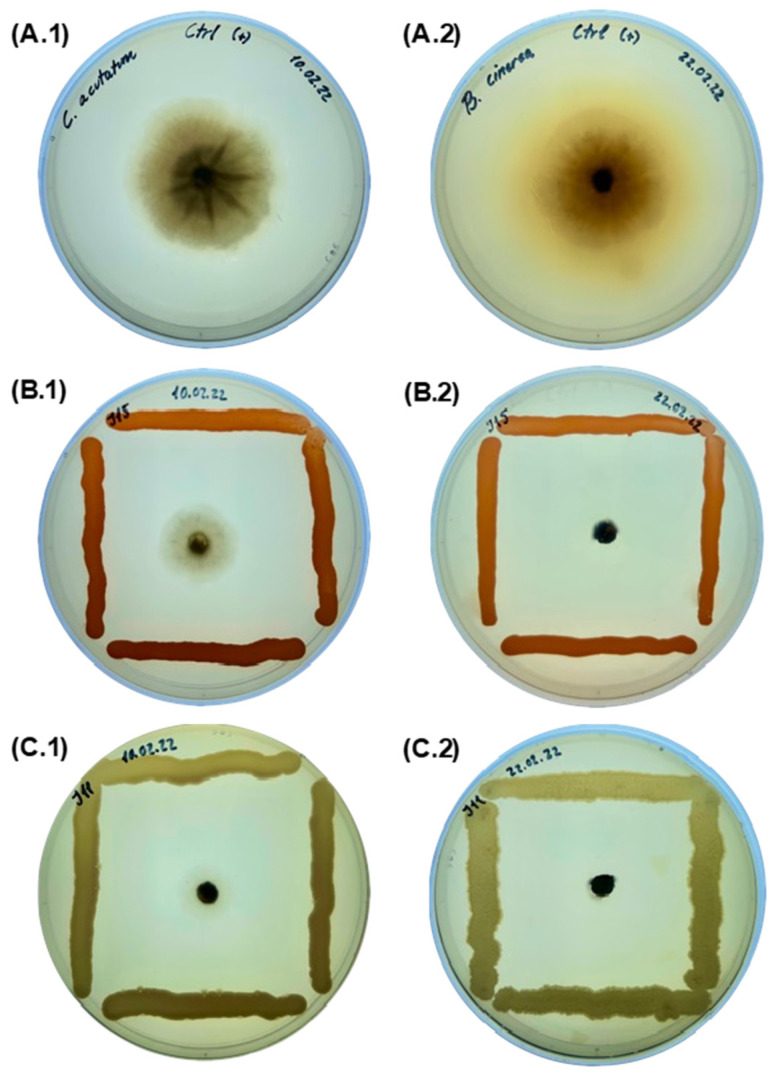
Inhibitory effect of ginger endophytic bacteria against *C. acutatum* IMI34849 (**1**) and *B. cinerea* B05.10 (**2**) in dual culture on LB medium after 7 days of incubation at 25 °C. (**A**) Pure culture of fungi (control +); (**B**) dual culture of *Kocuria polaris* J15 strain against fungi; (**C**) dual culture of *Lelliottia amnigena* J11 strain against fungi. Image captured with a camera system using a wide-angle lens with a focal length of 26 mm and an aperture of f/1.6.

**Figure 3 plants-12-04032-f003:**
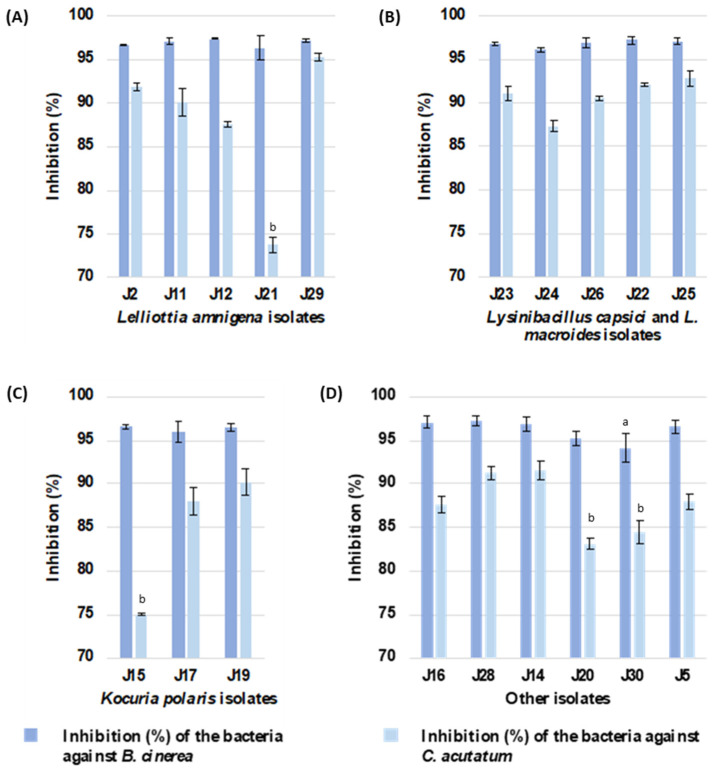
Inhibition (%) of the growth of *B. cinerea* B05.10 and *C. acutatum* IMI34849 against strains of endophytic *Z. officinale* at 7 dpi. Data are shown as mean of 3 replicates ± standard deviation. (**A**) Growth inhibition by isolates of *Lelliottia amnigena* species. (**B**) Growth inhibition by isolates of *Lysinibacillus capsici* (J23, J24, and J26) and *Lysinibacillus macroides* (J22 and J25) species. (**C**) Growth inhibition by isolates of *Kocuria polaris* species. (**D**) Growth inhibition by isolates of *Agrococcus citreus* (J16 and J28), *Agrobacterium tumefaciens* (J14), *Zymobacter palmae* (J20), *Acinetobacter schindleri* (J30), and *Mycolicibacterium* sp. (J5) species. The letters indicate that the in vitro inhibition produced by the isolate is significantly different from that produced by the other isolates, according to one-way ANOVA, Tukey’s post hoc test (*p*-value < 0.05).

**Figure 4 plants-12-04032-f004:**
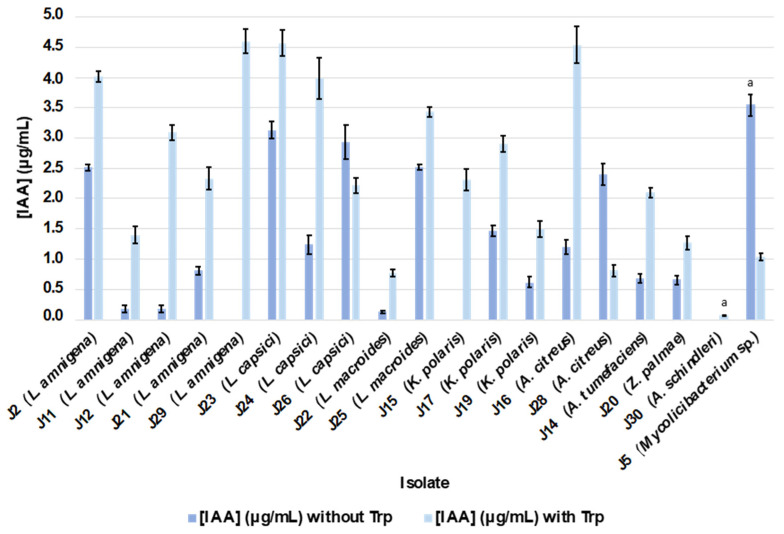
Quantification of indole acetic acid (IAA, µg/mL) produced by isolates in the presence and absence of the amino acid tryptophan (Trp). Data represent the mean ± standard deviation. The letter indicates that the amount of IAA produced by the isolate is significantly different from that produced by the other isolates, according to one-way ANOVA, Tukey’s post hoc test (*p*-value < 0.05).

**Table 1 plants-12-04032-t001:** Molecular identification of endophytic bacterial isolates from ginger.

Isolate	Identification	GenBank Acc. N.	
		16S	*rpo*B
J2	*Lelliottia amnigena*	OR840641	OR879032
J5	*Mycolicibacterium* sp.	OR840642	OR879039
J11	*Lelliottia amnigena*	OR840643	OR879034
J12	*Lelliottia amnigena*	OR840644	OR879035
J14	*Agrobacterium tumefaciens*	OR840645	OR879040
J15	*Kocuria polaris*	OR840646	OR879029
J16	*Agrococcus citreus*	OR840647	OR879037
J17	*Kocuria polaris*	OR840648	OR879030
J19	*Kocuria polaris*	OR840649	OR879031
J20	*Zymobacter palmae*	OR840650	OR879041
J21	*Lelliottia amnigena*	OR840651	OR879033
J22	*Lysinibacillus macroides*	OR840652	OR879027
J23	*Lysinibacillus capsici*	OR840653	OR879024
J24	*Lysinibacillus capsici*	OR840654	OR879025
J25	*Lysinibacillus macroides*	OR840655	OR879028
J26	*Lysinibacillus capsici*	OR840656	OR879026
J28	*Agrococcus citreus*	OR840657	OR879038
J29	*Lelliottia amnigena*	OR840658	OR879036
J30	*Acinetobacter schindleri*	OR840659	OR879042

**Table 2 plants-12-04032-t002:** Detection of lipopeptide production-related genes in endophytic bacteria via specific primer PCR.

Isolate	Identification	Bacylisin	Bacillomycin	Fengycin	Iturin	Subtilin	Surfactin
J2	*Lelliottia amnigena*	-	+	+	+	-	+
J11		-	-	-	-	-	-
J12		-	-	-	-	+	+
J21		-	-	-	-	+	+
J29		-	-	-	-	+	+
J23	*Lysinibacillus capsici*	-	-	-	-	+	+
J24		-	-	-	-	-	+
J26		-	-	-	-	+	+
J22	*Lysinibacillus macroides*	+	-	-	+	-	+
J25		-	-	-	-	-	+
J15	*Kocuria polaris*	-	-	-	-	-	-
J17		-	-	-	-	-	+
J19		-	-	-	-	-	+
J16	*Agrococcus citreus*	-	-	-	-	-	-
J28		-	-	-	-	+	+
J14	*Agrobacterium tumefaciens*	-	+	+	+	-	+
J20	*Zymobacter palmae*	-	+	-	-	+	+
J30	*Acinetobacter schindleri*	-	-	-	-	+	-
J5	*Mycolicibacterium* sp.	-	-	-	-	+	+

The + symbol indicates genes that were amplified via PCR, while the - symbol indicates genes that were not amplified via PCR.

**Table 3 plants-12-04032-t003:** Assessment of extracellular enzyme production by the 19 ginger endophytic bacteria following incubation at 25 °C on plates with specific culture media.

Isolate	Identification	Amylase	Cellulase	Esterase	Lipase	Protease
J2	*Lelliottia amnigena*	-	-	-	-	+
J11		-	-	-	-	-
J12		-	-	-	-	+
J21		-	-	+	+	+
J29		-	-	+	+	-
J23	*Lysinibacillus capsici*	-	-	+	+	-
J24		-	-	-	-	+
J26		-	-	+	+	-
J22	*Lysinibacillus macroides*	-	-	+	+	+
J25		-	-	-	-	+
J15	*Kocuria polaris*	-	-	+	+	-
J17		-	-	-	-	+
J19		-	-	-	-	+
J16	*Agrococcus citreus*	-	-	-	-	-
J28		-	-	+	+	-
J14	*Agrobacterium tumefaciens*	-	-	+	+	-
J20	*Zymobacter palmae*	-	-	-	+	+
J30	*Acinetobacter schindleri*	-	-	-	+	-
J5	*Mycolicibacterium* sp.	-	-	-	+	+

The + symbol indicates that a zone of clear hydrolysis was observed around the inoculum in the amylase, cellulase, and protease tests and that a zone of precipitation was observed in the esterase and lipase tests. The - symbol indicates that no zone of hydrolysis was observed around the inoculum in the amylase, cellulase, and protease tests and that no zone of precipitation was observed in the esterase and lipase tests.

**Table 4 plants-12-04032-t004:** Plant-growth-promoting potential of isolates.

Isolate	Identification	Fixation of Atmospheric Nitrogen	Solubilization of Potassium	Solubilization of Phosphate	Production of Siderophores	Production of IAA (without Trp/with Trp)	Production of Biofilm
J2	*Lelliottia amnigena*	+	-	-	-	+/+	+
J11		-	-	-	-	+/+	+
J12		-	-	-	+	+/+	+
J21		-	-	-	-	+/+	+
J29		-	+	+	-	-/+	+
J23	*Lysinibacillus capsici*	-	-	-	-	+/+	-
J24		+	-	-	-	+/+	+
J26		+	-	-	+	+/+	+
J22	*Lysinibacillus macroides*	-	-	-	-	+/+	-
J25		+	-	-	-	+/+	+
J15	*Kocuria polaris*	-	-	-	-	-/+	-
J17		+	-	-	-	+/+	+
J19		+	-	-	-	+/+	+
J16	*Agrococcus citreus*	+	-	-	-	+/+	+
J28		+	-	-	+	+/+	+
J14	*Agrobacterium tumefaciens*	-	-	-	-	+/+	+
J20	*Zymobacter palmae*	-	-	-	-	+/+	-
J30	*Acinetobacter schindleri*	-	-	-	-	-/+	-
J5	*Mycolicibacterium* sp.	+	-	-	+	+/+	+

IAA, indole acetic acid determinant without and with the addition of tryptophan (Trp) as a precursor. The + symbol indicates a positive reaction or the presence of the compound, while the - symbol indicates a negative reaction or the absence of the compound.

**Table 5 plants-12-04032-t005:** Classification of bacteria based on criteria described by Stepanovic et al. [24].

Strongly Adherent	Moderately Adherent	Weakly Adherent	Not Adherent
*Mycolicibacterium* sp. J5 *K. polaris* J19 *L. capsici* J26 *A. citreus* J28	*L. amnigena* J2, 12 *K. polaris* J17 *L. macroides* J25	*L. amnigena* J11, J21, J29 *A. tumefaciens* J14 *A. citreus* J16 *L. capsici* J24	*L. macroides* J22 *L. capsici* J23 *K. polaris* J15 *Z. palmae* J20 *A. schindleri* J30

**Table 6 plants-12-04032-t006:** Specific primers used to amplify the *16S rRNA* gene and *rpo*β gene.

Primer	Sequence (5′ → 3′)	Target
*16S_F*	GAAGAGTTTGATCATGGCTC	*16S* rRNA gene [81]
*16S_R*	AAGGAGGTGATCCAGCCGCA	
*Univ_rpo*β*_F_deg*	GGYTWYGAAGTNCGHGACGTDCA	*rpo*β gene [82]
*Univ_rpo*β*_R_deg*	TGACGYTGCATGTTBGMRCATMA	

**Table 7 plants-12-04032-t007:** Primers used for identification of genes involved in the lipopeptide synthesis.

Primer	Sequence (5′ → 3′)	Product Size (bp)	Used for
*BACF*	CAGCTCATGGGAATGCTTTT	500	Detection of *bac*A gene (bacylisin)
*BACR*	CTCGGTCCTGAAGGGACAAG		
*BMYBF*	GAATCCCGTTGTTCTCCAAA	370	Detection of *bmy*B gene (bacillomycin)
*BMYBR*	GCGGGTATTGAATGCTTGTT		
*FENDF*	GGCCCGTTCTCTAAATCCAT	270	Detection of *fen*D gene (fengycin)
*FENDR*	GTCATGCTGACGAGAGCAAA		
*ITUCF*	GGCTGCTGCAGATGCTTTAT	423	Detection of *itu*C gene (iturin)
*ITUCR*	TCGCAGATAATCGCAGTGAG		
*SPASF*	GGTTTGTTGGATGGAGCTGT	375	Detection of *spa*S gene (subtilin)
*SPASR*	GCAAGGAGTCAGAGCAAGGT		
*SRFAF*	TCGGGACAGGAAGACATCAT	200	Detection of *sfr*AA gene (surfactin)
*SRFAR*	CCACTCAAACGGATAATCCTGA		

## Data Availability

Data are contained within the article and Appendix A.

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
