# Peer review of "In Vitro Studies of Endophytic Bacteria Isolated from Ginger (*Zingiber officinale*) as Potential Plant-Growth-Promoting and Biocontrol Agents against *Botrytis cinerea* and *Colletotrichum acutatum"

_plants, 2023, doi:10.3390/plants12234032_

Round 1

Reviewer 1 Report

Comments and Suggestions for Authors

This manuscript describes the isolation of endophytic bacteria from the rhizomes of ginger, with the aim of identifying and describing phenotypes with potential for the biocontrol of fungal diseases caused by phytopathogenic fungi (B. cinerea and C. acutatum) and for plant growth promotion. Nineteen isolates were obtained, which were identified using molecular methods, and phenotypically characterized using molecular and biochemical methods. All bacterial strains exhibited in vitro inhibitory activity against the two fungi, and most possessed a varying array of metabolic potential to produce different chemicals that may be involved in antibiosis. Plant growth promotion potential was assessed by phenotypic determinations of metabolic capacity. The authors conclude that many of the ginger endophytes obtained have the potential to act as biocontrol and/or plant growth promotion bacteria. Overall, this is a competently performed study, if quite preliminary, which stops short of demonstrating actual utility of any of the isolates. Nevertheless, this is suitably acknowledged by the authors and the work is still interesting and useful. I have made several minor suggestions for improvement, most importantly in the rigor with which the strains have been identified to the genus or species level. These are detailed below.

1.     Line 82: Rather than, “subway rhizome”, perhaps the authors mean, “subterranean rhizome" - please check and correct.

2.     Lines 96-106: Nucleotide sequence similarities alone are insufficient for taxonomic identification, since multiple species can have identical percent identities. The authors must perform phylogenetic analysis of the sequences generated from the isolates, for both 16S and rpoB. This is essential for the rigor of taxonomic identification.\

3.     Table 1: It is not clear which sequences are described here - 16S or rpoB? Both? No phylogenetic analysis was performed. The accession numbers provided refer to BLAST hit, not to the actual sequences determined, which do not seem to have accession numbers. These must be deposited to a publicly available repository and accession numbers provided.

4.     On what basis were the isolates assumed to be identified to the species level? The criteria used need to be specified. Moreover, in the absence of phylogenetic analysis, BLAST-determined percent identities are insufficient for taxonomic identification.

5.     Line 114: The presence of the genes does not necessarily indicate that the strains actually produce these metabolites. This is only a potential, and the authors should clearly acknowledge this.

6.     Figure 1: This gel looks artificial, and the legend indicates, "schematic". If these are not the actual PCR products, what is the point of presenting this as a Figure? It should at least represent a real gel - if not, a table indicating presence of absence of a band of appropriate sizes would suffice.

7.     Lines 245-266: Related to points 2 and 4 above, the taxonomic identification of the isolates has not been demonstrated with sufficient rigor without phylogenetic analysis. This paragraph may have to change once a suitable analysis is performed.

8.     Line 486: presumably the authors mean, “cellulase” activity – please correct typographical error.

Comments on the Quality of English Language

only minor issues detected - eg "within the agricultural and food,...". Nothing that impedes the understanding of the material.

Author Response

Attached is the detailed response to each of the comments made by the reviewer.

Reviewer 2 Report

Comments and Suggestions for Authors

General comments:

This is an interesting manuscript presenting the isolation identification and phenotypic evaluation of ginger rhizome endophytes. The introduction provide all the necessary background information for the presented experimental part. The results are correctly presented. The authors discuss their findings with published data drawing meaningful conclusions. The methods are well selected and mostly adequately described. Also the conclusions and selected literature is fitting. I have but a few comments upon which the manuscript could be improved.

Firstly I recommend that the authors should include the supplementary data to their manuscript and put their all the data used to prepare figures, and sequences used for the determination of species or genus of isolated strains.  Additionally figure and table captions should include more data. The Figure/table caption should be sufficient to fully understand the data presented in the figure/table. The authors should also add the disclaimer at the bottom of the references and provide uncropped version of the gel photography used for figure 1.

In general I consider this manuscript as acceptable for publication after minor revisions.

Please find my detailed comments bellow:

Please add Ginger to the title and put scientific name in brackets.

Line 21 please add the descriptor abbreviation to the scientific names of plats (Rosc.)

Line 102 I could count 9 different species (2 Lysinibacillus) please check this.

Line 108: Based on the present results you can say that you could detect the presence of genes responsible for the production of certain antimicrobials, not the production by itself. Some of the antimicrobials might be produced only in specific circumstances. Detection or no detection of genes does not automatically imply production or no-production.

Line 132: Please provide the growth medium used for the test

Line 143: Please add the information on the used method for the detection of this metabolites to the table caption.

Line 151: Please add the information on the used medium, temperature of incubation, the number of technical and biological repetitions and if possible the statistics. Additionally please explain how the inhibition was calculated. The figure caption should be sufficient to interpret the data presented on the figure. Please change the font color in the figures to black to increase visibility.

Line 165: Please write scientific names in italics in this paragraph.

Line 176: Please shortly explain how such activities were verified.

Line 182: None of the tested abilities.

Line 183: Please decrease the font size of the table header.

Line 206: Please use black or larger font for this figure, include statistics if possible and add the number of repetition to the figure caption. Additionally please use dot (.) as a decimal separator and with instead of con. And without instead of sin.

210: biofilm

Line 248: Please add English names and descriptor abbreviation.

Line 262: Please add the scientific name.

Line 369: Biofilm

Line 463: Was the test performed on LA plates? What was the growth medium for isolates growth before the experiment. And what medium was used for fungal propagation. What was the volume and density of the bacteria used, or were they incubated from solid medium.

Line 474: Please provide rcf’s

Line 525 Rcf and use degree symbol:°

Line 608: Please add the supplementary materials: Data used for figure preparation in table format, the sequences used for the isolates assignment to genus and species.

Author Response

(The authors gave the same response as above.)

Reviewer 3 Report

Comments and Suggestions for Authors

Informative and well structured article. Emphasized, perhaps somewhat excessively, the possible use of endophytes.

 Some minor corrections:

1)     Line 33 – Write all Latin names in italics

2)     Table 1 – 98.63; 85.96 etc.

3)     Line 137 – Round 95.3% to 95 as in other cases

4)     Chapter 2.4.1. – Write all Latin names in italics

5)     Line 180 – “and Mycolicibacterium” instead of “and Mycolicibacterium

6)     Add reference to table 4 2.4.

7)     Lines 210-212 – Correct the first sentence!

8)     Line 254 – Why “simple plants”?

9)     Line 292 – Colletotrichum gloeosporioides

10) Line 486 – should be cellulase activity

11) Line 492 – should be CaCl2·2H2O

12) Line 511 – pH 7.0-7.5

13) Line 538 – 530 nm

14) Lines 574-589 – It is unnecessary, if only to shorten it to 1-2 sentences.

Author Response

(The authors gave the same response as above.)
